# Text Mining Approaches to Language Use in Social Media: The Case of Portuguese *Bué*

Camila Lívio [1,*,†] and Chad Howe [2,†]

1    Department of Research and Computational Data Management, University of Georgia Libraries, Athens, GA 30602-1641, USA
2    Department of Romance Languages, University of Georgia, Athens, GA 30602-1815, USA; chowe@uga.edu
*    Correspondence: camila.emidio25@uga.edu
†    These authors contributed equally to this work.

**Abstract:** This study describes processes of language change in Angolan Portuguese focusing on the use of intensifiers. Previous studies have shown that intensifiers are a relevant category for the study of language change due to their rapid change and variable meaning. It has been noted that intensifiers are particularly prone to renewal, suggesting speakers' desire to innovate. Informed by a Digital Humanities approach, we collect and analyze data from Twitter (now X), focusing on the multi-functional intensifier *bué*, 'very', in Angolan Portuguese (AP). In this paper, we (1) provide an overview of the word's distribution in AP, (2) consider the processes of change involved in *bué*'s variation, and (3) discuss the role of linguistic borrowing in language change and grammaticalization, shedding light on some of the cultural aspects that play a role in this word's development, such as the influence of the media and the contact situation between Angolan and European Portuguese.

**Keywords:** intensifiers; Angolan Portuguese; European Portuguese; social media; text mining; language variation and change

## 1. Introduction

Linguistic innovation and intensifiers are concepts that often appear together. The relationship between them can be described through the Maxim of Extravagance proposed by Haspelmath (1999, p. 1055): "talk in such a way that you are noticed". Intensifiers, also known as degree modifiers, amplifiers, maximizers, and boosters (Tagliamonte 2008, p. 361), are argued to be a highly fluctuating class (Peters 1994, p. 271). This fluctuation is the result of the competition among intensifying forms, which change and are recycled in speech with renewed meaning (Ito and Tagliamonte 2003; Macaulay 2006; Tagliamonte 2008; Kanwit et al. 2017). For example, Tagliamonte observes that the low rate of *very* in Toronto confirms the interpretation of the preference for *really* over *very* in the twentieth century (Tagliamonte 2008, p. 369). Such rapid renewal in the intensifying system leads to the argument that speakers seek expressive (or extravagant) ways to emphasize meaning, "since their impact is only as good as their novelty" (Tagliamonte 2008, p. 391). This leads us to reflect upon issues of agency in language. As far as intensifiers are concerned, the adoption of a newer form (e.g., *wicked*) in the face of more canonical ones (e.g., *very*) can be conceptualized in terms of accommodation (Backus and Spotti 2012, p. 190). In other words, speakers tend to—consciously or unconsciously—adapt to various social contexts and the impressions they wish to make on their interlocutors (Nguyen et al. 2016, p. 550).

In this paper, rather than discussing a novel intensifier, we analyze the distribution, usage, and collocational profile of the well-established intensifier *bué*, 'very', in Angolan Portuguese. The origins of the word are not completely clear; several sources indicate that it comes from Kimbundu, a Bantu language widely spoken in Angola (1,700,160 speakers) and with an ethnic population of 6,000,000 people (Campbell 2008). For instance, in the online Portuguese dictionary *Priberam* (Figueira et al. 2011), the word appears as a synonym

of *muito* in the informal Portuguese spoken in Portugal. The authors of the dictionary entry affirm that the Kimbundu origin is, however, uncertain. Pimenta (2022, p. 64) affirms that the use of this word was, for the generations born from the mid-1970s onwards, associated with migrating Angolans living in Portugal or with Angolan roots. Pimenta (2022) also refers to them as the *Bué Generation*, a generation with an "urgency for existence" (Pimenta 2022, pp. 64–65), and suggests that this generation was responsible for adopting the word from Kimbundu, *mbuwe*, into Portuguese, adding that the word's consolidation as a linguistic mark of identity of Lisbon's suburban culture stems from a song by Portuguese rapper of Cape Verdean roots, Boss AC (Pimenta 2022, p. 64). In addition, Almeida (2008, p. 19) mentions the frequent use of the word as an intensifier in the teen Portuguese soap opera *Morangos com Açúcar*, which aired between 2003 and 2012. Almeida (2008) explains that her choice to investigate the soap opera as a corpus for her study stems from the fact that such creative work featured 'youngspeech,' given its audience. Hence, the frequent use of *bué* by this age group is considered evidence of the word's adoption into European Portuguese. A similar kind of media influence on language use was documented by Tagliamonte and Roberts (2005), in which the authors argue that the sitcom *Friends* has been "an influential cultural phenomenon" that "provides a kind of preview of mainstream language" (Tagliamonte and Roberts 2005, p. 296). The authors affirm that these types of media seem to "pave the way", as language tends to be more innovative in the media than in the general population (Tagliamonte and Roberts 2005, p. 296). While the influence of television and other media has often been overlooked as a relevant factor in language change within the field of sociolinguistics (Stuart-Smith 2007, p. 140), we consider that media, such as television and music, serves as an indirect catalyst for linguistic diffusion. As Stuart-Smith (2007) puts it, "television may act as a source for new lexis and idioms, or as a model for speakers of a dialect to acquire the core phonology and syntax of the standard variety of a language (. . .), but here such changes require conscious motivation by speakers to orientate towards, and imitate, such a model" (Stuart-Smith 2007, p. 140). Trudgill (1989, p. 228) affirms that the degree of contact among speakers of different varieties of a language influences the rate of change. Therefore, considering the context of contact between Angolans and Portuguese is meaningful for a more comprehensive understanding of *bué*'s development.

During the Portuguese expansion, from the sixteenth to the twentieth century, speakers came into contact with a variety of different communities and languages. Specifically in the case of Angola, Carvalho and Lucchesi (2016, p. 44) affirm that the dissemination of Portuguese was limited even when the Portuguese ruled Luanda, in the late fifteenth century. The language was mostly used by the African–Portuguese administrative elite as a second language until the mid-eighteenth century (Carvalho and Lucchesi 2016, pp. 44–45). This linguistic situation changed in the twentieth century when a new policy requiring that Angolans were fluent in Portuguese was implemented during the Salazar dictatorship (1928–1974), and even then, the proportion of Portuguese speakers in Angola amounted to only 1% by 1940, dropping again by the time it reached political independence in 1975 (Carvalho and Lucchesi 2016, p. 44). Carvalho and Lucchesi (2016) add that in the present day, only a small portion of the Angolan elite speaks Portuguese as their mother language, in a variety that is similar to European Portuguese, as more than 90 percent of the population is a native speaker of a Bantu language, such as Kimbundo, Kikongo, and Umbundo (Carvalho and Lucchesi 2016, p. 45). A wide influx of immigrants from Angola to Portugal took place in the 1970s. An estimated 500,000 to 800,000 Portuguese settlers known as *retornados*—a term used to describe the Portuguese who were born in the African colonies—were forced to return to Portugal during the decolonization (Peralta 2022, p. 54). This exchange between Portugal and Angola is not unidirectional. With the end of the Civil War, many Angolans who used to live in Portugal returned to Angola, as well as new generations of Angolans moved to the former colony, which resulted in a steady growth of this population, according to the Portuguese National Institute of Statistics (INE 2022). Oliveira (2005) affirms that the relations between Angola and Portugal have been marked

by the dynamics of hierarchy and silencing since the fifteenth century, which resulted in great economic, social, cultural, and linguistic changes in Angola and, to a lesser degree, Portugal (Oliveira 2005, pp. 56–57).

This intense contact has, not surprisingly, led to a number of linguistic changes. Carvalho and Lucchesi (2016, p. 42) maintain that in most cases, the effects of language contact are observed through lexical borrowings that do not directly affect the languages involved. This claim seems to apply to our case study since the use of the word *bué* in Portugal tells us a small part of the long-standing relationship between these countries.

Our synchronic analysis utilizes a corpus of geo-tagged tweets from Luanda, Angola, and uses a computational approach to text analysis. Considering the nature of our dataset, we focus on language-internal factors that contribute to the variation and change of *bué*. We find this intensifier to be a compelling form for quantitative analysis of language variation and change due to its (i) development and variable use and (ii) frequency of use. Furthermore, it provides evidence for arguments that have been made in the study of intensifiers in English, such as the notion that intensifiers are considered to be (1) constantly changing (Ito and Tagliamonte 2003; Méndez-Naya 2003), (2) age-related (Roels and Enghels 2020), and, as we contend, (3) prone to being borrowed to/from other languages. Examples (1), (2), and (3) were taken from *Corpus do Português:Web/Dialects* (Davies and Ferreira 2006) in the Angolan subcorpus.

(1)   *Meus amigos, não adianta só **filosofar bué***
      'My friends, there's no point in philosophizing a lot'
(2)   *Foi uma cena **bué positiva** pois tive a oportunidade de mostrar outros trabalhos meus...*
      'It was a very positive experience because I had the opportunity to show some of my other work'
(3)   ***bué de** peeps no mambo, boa música e BAR ABERTO!!! Daammm*
      'Lots of folks dancing, good music and OPEN BAR!!! Dammm'

We start by outlining an overview of the literature on intensifiers in general and on Portuguese specifically, followed by a description of our materials and methods. We then show the general trends of intensification in the Twitter corpus, which are relevant to the introduction of our research questions in Section 6. The results are divided into three parts: (1) the general distribution of *bué* in the corpus, (2) a trigram-based approach for the analysis of the right and left collocates of *bué*, and (3) a statistical analysis of association strength. We then provide a discussion of the results and a conclusion in Section 7.

## 2. An Overview of Intensifiers

Speakers use a wide range of strategies to ensure that they, in Haspelmath's words, are noticed. These include types of linguistic structures that intensify, or "boost", the meaning of, among other categories, adjectival modifiers (Tagliamonte 2008)[1]. For instance, the example in (4) compares the modifiers *alto/tall* and *muito alto/very tall*. The effect of the lexical intensifiers *muito* and *very* is such that the hearer is expected to understand that Cris is taller than Carmo. This interpretation holds regardless of whether the propositions are presented in parallel, as in (4a), or contrasted, as in (4b). The use of items like *muito* and "very" as adjectival modifiers will constitute the focus of the current analysis, although, as will be presented later, these effects are obtained in other domains as well. Among the other structures used with an intensifier meaning is the morphological superlative *-íssimo*, illustrated in (4c). Although this structure can be used with a meaning akin to that of *muito*, we will set aside a more in-depth discussion of how superlatives might be integrated into the system of degree modification in Portuguese.

(4)   a.   *Carmo é alto e Cris é **muito alto**.*
           'Carmo is tall and Cris is **very tall**.'
      b.   *Carmo é alto mas Cris é **muito alto**.*
           'Carmo is tall but Cris is **very tall**.'
      c.   *Carmo é alto mas Cris é **altíssimo**.*
           'Carmo is tall but Cris is **very tall/the tallest**.'

An exhaustive discussion of the precise semantic and pragmatic details surrounding the use of lexical intensifiers will not be provided in this discussion, though it should be noted that, at least for the purposes of explaining semantic change, it is precisely the comparison between, on the one hand, unmodified adjectives and those whose meanings are "boosted" (as in 4), and, on the other, the interplay between different lexical items drafted into service by speakers in this domain. The variety of lexical intensifiers in English has been documented in numerous sources (Bolinger 1972; Ito and Tagliamonte 2003; Macaulay 2006; Tagliamonte 2008, among others), with two primary observations that stand out in the literature. First, the individual entries in this set—such as *really*, *very*, *pretty*, *wicked*, etc.—emerge in sequence diachronically, with older variants existing alongside or perhaps being replaced with newer ones. This means that, for any given point in synchrony, speakers have at their disposal a palette of intensifiers that can be used for pragmatic purposes (i.e., "boosting" or "maximizing"). The second, and related, observation specifically concerns this notion of replacement, or "recycling", as proposed by Ito and Tagliamonte (2003) and Tagliamonte (2008). The layering of options available to speakers produces competition among these structures, with some becoming more canonical, and consequently less "extravagant" (e.g., English *very*, than other, often newer options, like English *so* or *wicked*).

In Portuguese, Lívio and Howe (2020) argue, following Foltran and Nóbrega (2016), that there are two canonical intensifiers, *muito* and *bem*, exemplified in (5a) and (5b), respectively. These structures, discussed also by Gomes (2011), produce meanings analogous to their English counterparts, providing speakers with a means of enhancing the degree of the modified adjective. Foltran and Nóbrega (2016) also argue that *muito* is a "prototypical intensifier" in that there are numerous reflexes of this etymon in other Romance Languages[2].

(5)  a.  *Portanto, é **muito importante** que o usuário tenha disponível cópias de segurança recentes de seus dados.*
'Therefore, it is **very important** that the user has secure recent copies of their data.'
  b.  *E isto ensina-nos a terceira lição **bem importante**: a missão não é nossa.*
'And this is the third **really important** lesson we are taught: this is our mission.'

In their cross-dialectal distribution of *muito* and *bem*, Lívio and Howe (2020) show (i) that *muito* is preferred, at least in the context of adjectival modification, in each of the target varieties (i.e., Angolan, Brazilian, European, and Mozambican) and (ii) that there is some fluctuation with respect to this preference ranging from, on the high end, 87% *muito* in both the Angolan and Mozambican samples to only 70% in Brazil. These differences in overall frequency are accompanied by collocational distinctions as well, where *muito* co-occurs with a more restricted set of adjectives across the dialect samples (e.g., *simples* 'easy/simple', *bom* 'good', *difícil* 'difficult') and *bem* displaying a more heterogeneous profile.

Beyond these structures, Foltran and Nóbrega (2016) provide two additional groupings, "innovative adjectival intensifiers" (as in 6) and "borrowed intensifiers" (shown in 7 and 8). The use of *extremamente* in (6) reflects the typical trend of using adverbs in degree modification, a pattern attested widely across Portuguese and other Romance Languages. Examples (7) and (8), we argue, represent a distinct pattern concerning the development of these forms, suggesting an avenue for "renewal", following Tagliamonte's terminology, that draws on items from other languages. This is precisely the mechanism that is being proposed with *bué* in the current analysis.

(6)  *É um processo natural de floculação de baixo custo e **extremamente importante** para as regiões secas*
'It's a low-cost, natural flocculation process and **extremely important** for dry regions'
(7)  *Em o final, fiquei **super feliz** porque não contava mesmo com aquela classificação*
'In the end, I was **super happy** because I really didn't count on that classification'
(8)  *Eu tinha uma limitação muito grande na alimentação, nada de frituras, carne apenas de rã, uma alimentação **hiper controlada**.*
'I had a very big limitation in my diet, no fried foods, only frog meat, a **very controlled** diet.'

It perhaps goes without saying that a discussion of lexical borrowing of the type illustrated by the use of *super*, *hiper*, and *bué* in Portuguese necessitates an exploration of the cultural factors that influence their introduction. Nevertheless, we set aside these issues for now to focus on the structural embedding of these items, specifically *bué*, offering a detailed view of how it has been integrated into the system of degree modification. We will demonstrate that, in addition to its more canonical behavior as an adjective intensifier similar to both *muito* and *bem*, *bué* is distinct in a number of ways, including its usage in modification beyond the adjectival domain.

## 3. Materials and Methods

Recent studies have shown that Twitter data offer compelling linguistic representation in terms of regional tracking and diversity of the sample (Huang et al. 2016), as well as consisting of a useful source for the study of language variation and innovation (Grieve et al. 2018). Such an approach to linguistic analysis has been increasingly adopted as access to large corpora and texts become more available. Some emblematic examples include the works by Tagliamonte and colleagues on the investigation of intensifiers using corpora (Ito and Tagliamonte 2003; Tagliamonte 2008; Tagliamonte and Roberts 2005), Grieve and colleagues on the use Twitter as a prolific source of language data (Grieve et al. 2018, 2019), and Eisenstein and colleagues on social media language and its contribution and limitations to processing and understanding online natural language (Eisenstein 2013; Bamman et al. 2014). What these works have in common is that they center on different aspects of English and its variants. In Romance Languages, we observe a growing body of corpus-based studies that seek to not only apply similar methods to those used for the study of English in varying datasets but also deals with language-specific questions that go beyond English-oriented methods[3].

For the present study, we compiled a corpus of 10,000 geo-tagged tweets from Luanda, Angola, in September of 2021, using the scripting and software environment R (R Core Team 2013) and the package *Rtweet* (Kearney et al. 2016). Retweets were not included in the corpus as a way to prevent gathering repeated tweets and bots. Furthermore, tweets are mixed, specified by the argument "type". The search string targeted a list of five high-frequency adjectives (*bom* | *boa* 'good', *feliz* 'happy', *fixe* 'cool', *triste* 'sad', *especial* 'special') determined by a search on the Angolan Portuguese portion of the Reference Corpus of Contemporary Portuguese (do Nascimento et al. 2014). Thus, every tweet contains at least one of these adjectives specified in the search. The reasoning behind this methodological decision is that intensifiers are more frequently found modifying adjectives in comparison to other parts of speech (Ito and Tagliamonte 2003, p. 263). Further processing of the data included checking for its quality (i.e., number of empty values, duplicates, and consistency), cleaning (i.e., stripping the punctuation, case conversion, and removal of stop words), tokenization, and tagging. The tagging was performed using the R package *Udpipe* (Wijffels et al. 2018), due to its high performance and simplicity (Schweinberger 2023b). The package offers a great number of pre-trained language models, including Portuguese.

To carry out the analysis, we first provide overall trends in adjectival intensification in the Tweet corpus, followed by a manual search focusing on adjectives. We then focus on the structural properties of *bué* with the extraction and analysis of trigrams, as well as measuring the association strength between *bué* and other words. These methods allow for the observation of *bué*'s collocational profile, offering insights into the words' meaning and variation.

The extraction of linguistic samples from digital media sites has become a popular method allowing for the gathering of larger samples that do not involve any type of elicitation from participants, which is desirable in the sense that it enables "unobtrusive measurements" (Nguyen et al. 2020, p. 4). However, it is essential to point out some of the limitations that born-digital data entail. For example, Twitter data are not as widely available in the case of Angolan Portuguese in comparison with many varieties of English. The website *Statista* lists the top 20 countries whose users are more engaged in the mi-

croblogging platform, showing that the only Portuguese-speaking country that displays a high number of Twitter users currently is Brazil, with 24.03 million users (Statista 2023). For that reason, our sample was limited to 10,000 tweets. In addition, we agree with Morin and Grieve (2024) that a disadvantage of working with social media data is related to the generalization of the results. Twitter data can only provide insights into language use on Twitter and, consequently, about users from specific demographics that are more represented in this social media platform (Morin and Grieve 2024, p. 11).

*Data Management Plan*

Considering the principles of open data in linguistics, as discussed in (Berez-Kroeker 2022), we adhere to Mattern's guidelines (Mattern 2022, p. 19) on how to manage and preserve our dataset. While we strongly believe that a move toward open research (Gawne et al. 2021, p. 20) is imperative in our field to promote replicability and reproducibility, current regulations in the Twitter (now X) Developer Agreement prohibit full content redistribution, thereby affecting our ability to make the dataset entirely available. With these restrictions fully in mind, our data management plan, which outlines the data collection, software packages, dates, data structure, file types, metadata, variables, and analysis, can be found in this GitHub repository (commit efaf61c), https://github.com/camlivio/Bueh-DMP (accessed on 8 November 2023).

## 4. General Trends of Intensification in Tweets from Luanda, Angola

Initial mining of the corpus reveals the widespread use of intensifiers, such as *muito* ('very') and *bem* 'well', and comparative intensifying constructions, such as *mais ADJ que* ('more ADJ than') and *tão ADJ quanto* ('as ADJ as'), particularly *muito*. These findings align with the literature on intensification in modern Spanish and Portuguese, which highlights such trends (Kanwit et al. 2017; Foltran and Nóbrega 2016). Figure 1 illustrates such a trend by means of a rapid automatic keyword extraction (RAKE). Rose et al. (2010, pp. 3–4) explain that within the field of information retrieval, keywords are defined as a sequence of one or more words that represent a text's content. RAKE selects candidate keywords by focusing on meaning-bearing words and how they co-occur in the text. A score is then assigned to each candidate keyword and "defined as the sum of its member words scores" (Rose et al. 2010, p. 7). In other words, this method identifies keywords based on the frequency and the position of a word or set of words in a specific text[4]. Figure 1 shows the top 10 most frequent strings of adverbs preceding adjectives in the dataset, indicating that, as far as this syntactic combination goes, forms like *muito*, *tão*, and *bem* tend to be frequent before adjectives.

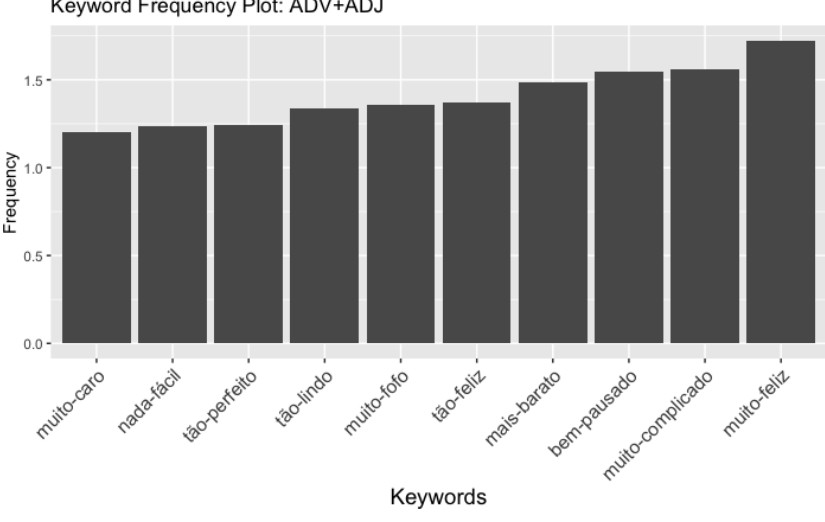

**Figure 1.** Rapid automatic keyword extraction of the sequence ADV + ADJ.

Lívio and Howe (2020, p. 480) show that the distribution of *muito* and *bem* preceding adjectives in two African Portuguese varieties, Angolan and Mozambican, is not parallel to that of Brazilian and European Portuguese, whose cases of *bem* + ADJ are significantly higher, especially in the Brazilian variety. Considering that *bem* is newer in comparison with *muito*—and given that the distribution of the latter is more homogeneous across varieties—we suggest that *bué* corresponds to *bem* in the Angolan variety. This can be better understood with a manual search in our Twitter corpus by adjective, as frequency shows us that *muito* and comparative forms *mais/que* and *tão/que* tend to be the most used intensifying forms before adjectives in the Angolan variety, as shown in Figure 1. We queried the corpus of Tweets to observe how and which morphemes are used in this morphosyntactic environment (i.e., adjectives modified by intensifiers), with the objective of exploring varying intensifiers regardless of their frequency. Table 1 features both modified high-frequency adjectives (*fixe* 'cool', *bom* 'good', *feliz* 'happy', *triste* 'sad') and adjectives that are highly affective but lower in frequency (*puto* 'shit', *cool*, *foda* 'screwed').

**Table 1.** Examples of intensified adjectives extracted from the Twitter corpus.

| Adjective | Example | Gloss |
|-----------|---------|-------|
| Fixe 'cool' | *tem sempre uma irmã bweeee fixe* | 'There's always a super cool sister' |
| Bom 'good' | *kkkk mto bom me sentir excluida to amando* | 'LOL so good to feel excluded I'm loving it' |
| Feliz 'happy' | *Aí tô mó feliz, comprei uma blusa* | 'Check it out I'm so happy, I got this new shirt' |
| Triste 'sad' | *estou tao triste mano queria bue reler aquela revista* | 'I'm so sad bro I really wanted to read that magazine' |
| Puto 'bitch' | *Cê tá onde, mó puto?* | 'Where are you/How are you feeling, so fed up?' |
| Cool | *O som está muito cool* | 'The music is really cool' |
| Foda 'shit' | *KKKKKKKK EU SOU MUITO FODA* | 'LOL I'm the shit' |

The concordances[5] in Table 1 show that intensifiers *bué*, also spelled *bue* and *bwé*, and *mó*—a written representation of the phonologically reduced *maior* ('bigger') (Xatara and Seco 2014, p. 511)—are the forms that appear to have a more informal use, similar to *bem*. We choose to focus on *bué* because it is the lexical intensifier with more language-specific characteristics and shows an interesting history and usage, illustrating why intensifiers have been considered to be (1) constantly changing (Ito and Tagliamonte 2003; Méndez-Naya 2003), (2) age-related (Roels and Enghels 2020), and (3) prone to being borrowed to and from other languages and communities.

As far as the language-specific argument is concerned, a quick search on *Corpus do Português* (Davies and Ferreira 2006) shows that *bué* is indeed widely employed in both Angolan and European Portuguese, as shown in Table 2. In this figure, both spellings are considered, *bué* and *bue*.

**Table 2.** Distribution of the intensifier *bué* | *bue* across varieties on *Corpus do Português Research Questions*.

| Word | All | Brazil | Portugal | Angola | Mozambique |
|------|-----|--------|----------|--------|------------|
| Bué | 1367 | 39 | 942 | 330 | 56 |
| Bue | 577 | 123 | 228 | 212 | 14 |

Considering the background provided in the previous sections, we are well-positioned to answer the following research questions:

RQ1: How is *bué* used as an intensifier in the dataset?

RQ2: Given the general distribution of the word in Angola, Brazil, Mozambique, and Portugal, how has *bué* become widespread among Portuguese speakers?

RQ3: What does *bué*'s collocational profile tell us about the path of change of intensifiers?

## 5. Results

### 5.1. Overall Distribution of Bué

To visualize how *bué* is used in the corpus, we compare it to the distribution of other lexical intensifiers—as opposed to intensifying affixes—such as *muito*, and alternative forms that are frequent in other variants of the language, like the use of *super* and *totalmente* in Brazilian Portuguese. We also show how the standard spelling of the word compares with alternative forms, such as *bue*, *bwe*, and *bued*. Tatman (2015, p. 97) argues that while it may seem counter-intuitive to look for phonetic information in written mediums such as Twitter, previous research has shown that "the parallels between face-to-face and computer mediated communication (CMC) are robust" and "use a high proportion of variant spellings" (Tatman 2015, p. 99). Spelling variation is relevant in that it reflects, at least to some extent, the speaker's own speech habits (Tatman 2015, pp. 99–100), and has been documented as a relevant way to observe social membership in some contexts (Androutsopoulos 2000; Thurlow and Brown 2003). We find that the word's spelling variation generates further data to observe *bué*'s various grammatical functions, particularly between intensifying and quantification constructions (e.g., *bué de* 'a lot of'). We provide a more detailed discussion about the many functions of the word in Section 6. Figure 2 illustrates these comparisons. In the first graph, only the form *bué* with the orthographic accent is considered.

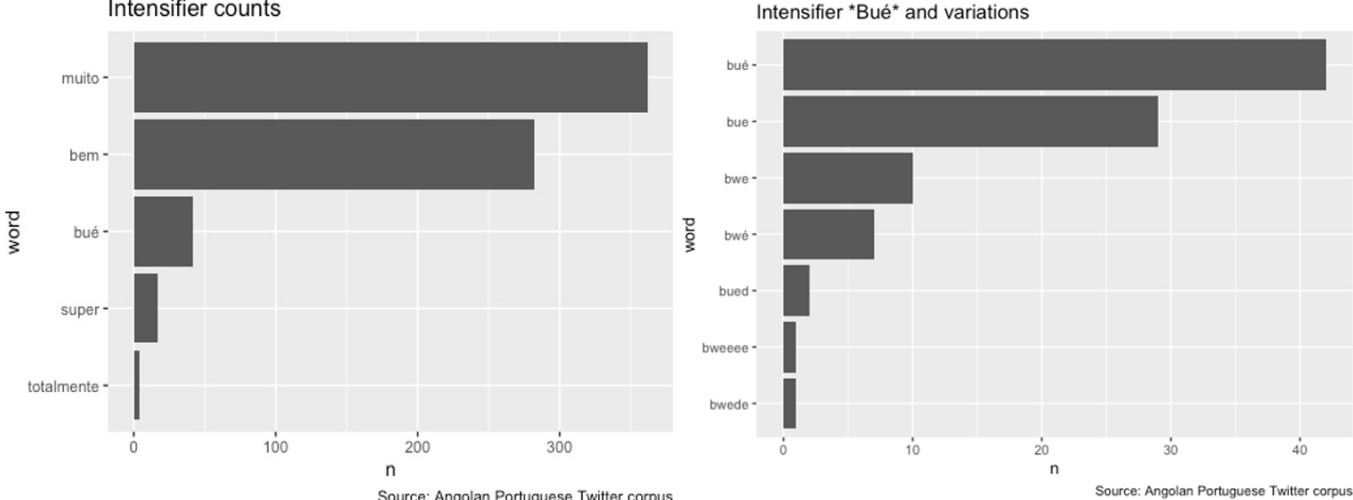

**Figure 2.** Comparing *Bué* to other lexical forms and variations in the same form.

It is interesting to notice that the overall counts of *bué*—even when considering only the normative spelling of the word—are higher than *super* since recent studies have shown that *super* has become widespread in many Portuguese varieties (Romerito Silva 2019; Lívio Emídio 2023), while significantly lower than both *muito* and *bem*. As far as the second graph in Figure 2 is concerned, we observe that both forms, with and without the accent, are the predominant choice among speakers in the sample. *Bued* and *bwede* point to a quantifying function of the word. Finally, Figure 3 shows the distribution of *bué* | *bue* by individual users. In this plot, users' handles have been anonymized and substituted for an identification number on the x-axis. The figure shows that a few users tend to use the word at higher rates than most users. The graph suggests that some are "super users" of the word, while most users tend to use it once[6].

**Figure 3.** Accounting for individual users.

*5.2. Trigrams*

In this section, we propose a corpus-based approach to understand the patterning and variability in expressions containing the string *bué*. Through the extraction of trigrams, defined as three-word sequences (Jurafsky and Martin 2023, p. 2), we observe the terms that most frequently appear before and after *bué*. For the present analysis, we first tokenized the Twitter corpus into trigrams, as illustrated below in Table 3, in which we observe the first ten lines of the output where word2 corresponds to *bué*[7].

**Table 3.** The first ten lines of trigrams containing *bué* in the word2 slot.

| id | word1 | word2 | word3 | Gloss |
|----|-------|-------|-------|-------|
| 1 | carne | bue | excited | 'meat very excited' |
| 2 | cena | bue | caricata | 'very ludicrous scene' |
| 3 | cena | bue | normal | 'very normal scene' |
| 4 | concordo | bue | mas | 'I totally agree but' |
| 5 | estao | bue | boas | 'they are so good' |
| 6 | estao | bue | nitidas | 'they are super clear' |
| 7 | fica | bue | bom | 'it's really good' |
| 8 | foi | bue | rough | 'it was really rough' |
| 9 | foi | bue | um | 'it was really one' |
| 10 | guardam | bue | de | 'they keep lots of' |

Essentially, this step results in a data frame in which each line contains a trigram. We then separate these trigrams so that each word is in its own, separate, column. With the inspection and extraction of all trigrams containing the word of interest, we created a new dataset in which *bué* occupies the second slot in the trigram, identified as *word2*. This arrangement facilitates the analysis of the words that precede and follow our key term. An illustration of a trigram featuring *bué* in the second slot is provided in Table 4.

**Table 4.** Representation of trigram extraction.

| Word1 | Word2 (=*bué*) | Word3 |
|---|---|---|
| *é* | *bué* | *fixe* |
| 'It's | very | cool' |

Jurafsky and Martin (2023, pp. 2–3) elaborate on the concept of n-grams, which rely on the conditional probability of a word *w* given its history *h*, formally denoted as $P(w|h)$. Essentially, the n-gram approach aids in addressing the question, "How frequently does the word *w* follow the history *h*?" (Jurafsky and Martin 2023, p. 2), thereby uncovering frequency trends in the co-occurrence of specific words. Our findings, illustrated in Figure 4, are crucial for the examination of the morphosyntactic environment of the term.

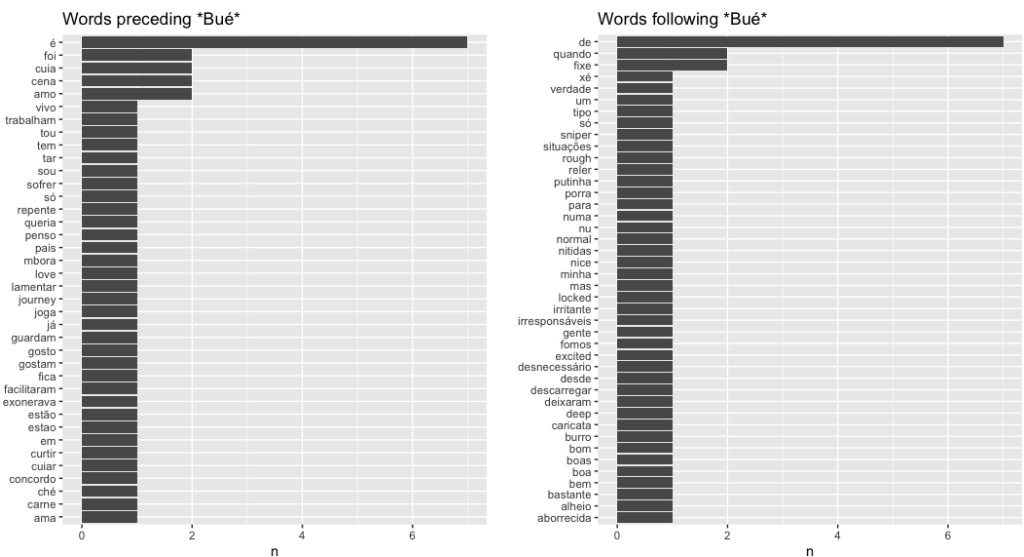

**Figure 4.** Words that precede and follow *bué*.

Starting with the first plot in Figure 4, the most frequent terms preceding *bué* is the singular form of the copular verb *é* 'is', followed by the less frequent preterit form of the same verb, *foi* 'was'. These forms indicate that speakers typically use *bué* as a modifier of predicative adjectives, such as the example given in Table 4. Tagliamonte (2008, p. 373) points out that a higher frequency of intensifiers modifying predicative adjectives is a reflection of "a later point in an intensifier's development", showcasing a well-documented pattern of intensification variation, namely, deslexicalization (Ito and Tagliamonte 2003; Tagliamonte 2008; Schweinberger 2020). This argument is motivated by the observation of the development of *very*: "According to some researchers, a later stage in the development of intensifier *very* was when it came to modify predicative adjectives" (Tagliamonte 2008, p. 373).

*5.3. Association Strength*

Corpus-based approaches have played a pivotal role in describing linguistic patterns of variation and use (Biber 2015, p. 197). The well-known phrase "You shall know a word by the company it keeps", attributed to Firth (1968, p. 179), remains pertinent in the context of our analysis. Specifically, we use an association measurement to shed light on *bué*'s collocates. We find this method to be useful in that the "use of computational tools ensures high reliability, since a computer program should make the same analytical decision every time it encounters the same linguistic phenomenon" (Biber 2015, p. 197). The trigram analysis allowed us to observe raw frequency counts of what terms occur as the right and

left collocates of *bué*. However, as Evert points out, plain frequencies are not enough "as a measure for the amount of 'glue' between two words" (Evert 2005, p. 20).

Our statistical interpretation of the frequency data is based on repeated Fisher's exact tests and subsequent Benjamini–Hochberg correction for repeated testing, following Schweinberger's (2023a) analysis of co-occurrences and collocations in R. The results indicate that the statistically significant collocations of *bué* in the Twitter data are *merdas* 'shit', *carne* 'meat', *tou* 'I am', and *morto* 'dead', as shown in the table below. The results in the column *CorrSignificance* represent the corrected p-values, which are necessary after performing several tests to avoid errors (Schweinberger 2023a). The corrected values display the only truly significant collocates of *bué | bue*. These findings hold practical value as they not only highlight the statistically significant collocates (*type*) but also the terms that are repelled (*antitype*) by the key term. The presence of the word *não*, as an antitype, shows that it co-occurs less frequently with the word *bué*.

| Term | CoocTerm | TermCoocFreq | CoocFreq | p | X2 | phi | expected | Significance | CorrSignificance | Type |
|------|----------|--------------|----------|---|-----|-----|----------|--------------|------------------|------|
| bue | merdas | 4 | 15 | 0.000193 | 30.4 | 0.24 | 0.345 | p < 0.001 | p < 0.01 | type |
| bue | carne | 4 | 22 | 0.000945 | 18.9 | 0.19 | 0.506 | p < 0.001 | p < 0.01 | type |
| bue | tou | 2 | 14 | 0.0378 | 4.5 | 0.09 | 0.322 | p < 0.05 | NA | type |
| bue | morto | 2 | 14 | 0.0378 | 4.5 | 0.09 | 0.322 | p < 0.05 | NA | type |
| bue | nao | 0 | 97 | 0.135 | 1.7 | 0.06 | 2.23 | NA | NA | antitype |
| bue | ver | 0 | 83 | 0.229 | 1.3 | 0.05 | 1.91 | NA | NA | antitype |
| bue | pq | 0 | 90 | 0.236 | 1.5 | 0.05 | 2.07 | NA | NA | antitype |
| bue | n | 0 | 62 | 0.377 | 0.7 | 0.04 | 1.43 | NA | NA | antitype |
| bue | bue | 0 | 41 | 0.612 | 0.2 | 0.02 | 0.943 | NA | NA | antitype |
| bue | acho | 0 | 35 | 1 | 0.1 | 0.02 | 0.805 | NA | NA | antitype |

## 6. Discussion

Breban and Davidse (2016, p. 245) and Méndez-Naya (2003, p. 389) argue that the various functions and layers of meaning of the intensifier *very* are observable due to its complete process of grammaticalization, which is the reason why *very* cannot be renewed (i.e., reintroduced in the speech with renewed expressive power as discussed in Tagliamonte (2008)). These authors, as well as several other studies on intensifiers (Ito and Tagliamonte 2003; Kanwit et al. 2017; Méndez-Naya 2003; Serradilla Castaño 2006), maintain that the high usage of an intensifier gives rise to alternative or competing forms in the language due to the speaker's desire for innovation and expressivity. Similar to *very*, *bué* has undergone many changes that are observable in our Twitter corpus, noticeably by its generalization of meaning (Méndez-Naya 2003, p. 372). For example, Almeida (2008) argues that the change is from positive adjective *bué* in Angola, as in examples (9) and (10), to intensifier *bué* in Portugal, as in (11).

(9) *A senhora é bué*
'The lady is fashionable'
(10) *Kota, o serviço foi bué, portanto vais dar só cem dólares*
'Kota, the service was good, therefore you should tip only $100'
(11) *Alunos estrangeiros, fixe. Vocês vão curtir bué a Barra*
'Foreign students, cool. You guys will enjoy a Barra a lot'

Adamson (2000, p. 47) describes a similar change in the adverb *lovely*. The author argues that *lovely* starts out as an adjective of positive connotation and comes to be used as an intensifier via a shift from denotational to a "speaker-related meaning", or from descriptive to affective, as well as a change that entails moving from being an independent lexical word to a grammatical operator (Breban and Davidse 2016, p. 222). The author contrasts the change in meaning and structure in *She is lovely* to *A lovely quiet engine* (Adamson 2000, p. 54), which is parallel to the process described by Almeida (2008) (*Ela é bué* to *é bué interessante*). To put it another way, our n-gram analysis of *bué* showed that the keyword tends to be preceded by copular verbs (e.g., *é* and *foi*), as well as other verbal forms (*amo* 'I love', *vivo* 'I live', *trabalham* 'they work', *tou* 'I am', *tem* 'have/has', *tar* 'are', *sou* 'I am', and *sofrer* 'to suffer'), indicating functional layering. This observation is further

supported when examining the right-hand collocates of *bué*, as the frequency in which the preposition *de* occurs points to the use of a quantifying construction. Examples (12) and (13) illustrate some instances of the construction in our Twitter corpus. Note that these examples display a "counting" interpretation (Doetjes 2008, p. 149) as the modifier targets the plural predicates.

(12)  *Bué de gajos*
      'Lots of guys'
(13)  *Bué de gatos*
      'Lots of cats'

As a quantifier, *bué de* displays similar behavior to the quantifying construction *monte de*, which is literally translated as 'a mountain of'. According to Alonso and Fumaux (2019, p. 141), the more recent quantifying interpretation of *um monte de* stems from a locative understanding of the construction. For instance, in example (14), from the 16th century, obtained by the authors from *Corpus do Português* (Davies and Ferreira 2006), they argue that the original construction is historically linked to a qualitative expression where *monte* ('mountain') refers to *morro* ('summit'), and the prepositional phrase establishes a connection to a specific location where the mountain is situated, as shown in example (14), taken from the historical section of *Corpus do Português* (Davies and Ferreira 2006) from the 16th century, as cited by Alonso and Fumaux (2019, pp. 141–42).

(14)  *Guar-te de praguejar de homës poderosos porque t oras hehû monte de africa. Onde foy enforcado daphitasgrãmatico porque dezia mal dos reys em verso*
      'Beware of cursing powerful men because you are but a mountain in Africa. Where Daphitas the grammarian was hanged because he spoke ill of kings in verse.'

Moreover, they add that such use functions as a specifier in that it identifies where the mountain is located. A number of syntagmatic variations are found in Portuguese, particularly as far as constructions of the type $NP_1$ *of* $NP_2$: *copo de vidro*, *caderno de atividades*, *monte de Lisboa*, as well as similar quantifying constructions, such as *milhares de pessoas* 'millions of people' and *centenas de problemas* 'hundreds of problems' (Alonso and Fumaux 2019, p. 122).

A further development of the quantifying construction is described by Almeida (2008, p. 122). The author states that *bué da* has evolved from *bué de* and was commonly used by young European Portuguese speakers to convey heightened expressiveness. This assertion is supported by examples (15) and (16) provided in her work (Almeida 2008, p. 118). By contrasting these examples with (12) and (13), we can observe that (15) and (16) produce an intensifying reading—rather than a counting one—despite the presence of the preposition *da*. This effect is attributed to the predicates they modify, as both *fixe* and *preocupado* are adjectives.

(15)  *Este jogo é bué da fixe eu ainda não joguei mas eu tenho o final fantasy x e é bué da fixe*
      'This game is super cool, I still haven't played it but I have Final Fantasy X and it is super cool'
(16)  *Acho que ficaste bué da preocupado*
      'I think you were super worried'

In addition to *bué de*, the results for the right-hand collocates display a variety of adjectives that are modified by the intensifier: *fixe* 'cool', *nítidas* 'clear', *nice*, *irritante* 'annoying', *irresponsáveis* 'irresponsible', *excited*, *burro* 'dumb', and *bom|boa* 'good'. This result 467 supports the claim that the more grammaticalized a form is, the wider the semantic evaluation of the words it modifies will be (Ito and Tagliamonte 2003; Kanwit et al. 2017). In other words, intensifiers that have been used by a community of speakers for a long time tend to modify adjectives of both positive and negative semantic evaluation, such as *bom* 'good' and *burro* 'dumb', as is the case with *muito* and, to a lesser extent, *bem* (*bem feio* 'well ugly'). Given that *bué* is no longer considered an innovative intensifier in Portuguese, evidenced by its inclusion in the *Dicionário da Academia das Ciências* (Casteleiro 2001), it is not surprising that speakers use it with a variety of adjectives.

The statistical analysis confirms that *bué* has undergone grammaticalization, aligning with the pattern found in similar constructions, such as *very* and *lovely*, and evidenced by the functional layers that it displays in the corpus (intensifier, quantifier, and adjective). Observing both the right-hand collocates of *bué* and its statistically significant collocates, we add that the type of intensification introduced by this word seems to be more pragmatic than grammatical in Beltrama and Bochnak's (2015) terms. The authors posit two main ways in which degree modification can manifest. The first, called "true degree intensification", occurs when an intensifier tracks scales lexically encoded in the modified word, as seen in examples like *very tall*, where "very" modifies a scalable property of the adjective "tall". In other words, they modify gradable predicates. Conversely, another class of degree modifiers targets aspects of the meaning rather than gradable properties, as in the Italian example *Michael Jordan è un campionissimo* 'Jordan is a super champion', in which the intensifying suffix *-issimo* reinforces the meaning of the modified word, enhancing the expressiveness of the utterance (Beltrama and Bochnak 2015, p. 845). The collocates in Figure 4 show that *bué* modifies both gradable (*fixe* 'cool'), and non-gradable predicates (*nu* 'naked' and *morto* 'dead'), though the true collocates of the word are non-gradable, suggesting that the interpretation of such constructions are context-dependent, or at least more dependent on the context than grammatical intensification.

In addition to showing the words with which *bué* is more likely to co-occur, an advantage of the code elaborated by Schweinberger (2023a) to measure word association is that we can concomitantly observe the terms that are repelled by our key term, *bué*. Negation words such as *não*, and its abbreviated form, *n*, are less likely to occur in contexts where *bué* is present. We suggest that at least part of the explanation for this phenomenon is connected to the issue of polarity. Similar to *bem*, *bué* partially retains its original meaning (Luo et al. 2019, pp. 1–2), which is a positive polarity meaning in this instance, conflicting with what Israel (2006, p. 10) describes as the "unpleasant sort of construction" that negation introduces.

In summary, we pose that *bué*'s "foreign quality" makes for an ideal intensifier as these modifiers are known for being "colorful" and "inventive" (Bolinger 1972, p. 18), and are often sourced from lexical items (Ghesquière and Davidse 2011, p. 252) as they are meaning-bearing words and speakers target some aspect of their meaning. For example, the English adjectives *awful* and *dead* function as intensifiers in utterances such as *an awfully long line* and *dead chuffed* due to their extreme meanings.

## 7. Conclusions

In this paper, we focused on the use of the Angolan Portuguese intensifier *bué* in social media data. We showed that as a synonym for 'abundance', and having its origins being generally attributed to Kimbundo, this frequent word in Angolan Portuguese displays many and variable uses. We argued that it is due to the intense contact among speakers of Angolan and European Portuguese, combined with its use in popular culture, that the word is borrowed into European Portuguese. Furthermore, we posit that, as a borrowing, *bué* possesses a foreign quality that can also be described in terms of Haspelmath's (1999) Maxim of Extravagance, or the Level of Surprisal that an intensifier introduces in speech. According to Scheffler et al. (2023, p. 2), "high surprisal of a sign means that it is unexpected given its preceding context of occurrence, and that it carries high information". In that sense, we believe that certain loanwords are apt to become thriving intensifying forms in a given language due to the high levels of information that they carry, explaining the popularity of other similar intensifiers categorized as foreign, such as *super* and *hiper* in many varieties of Portuguese (Foltran and Nóbrega 2016). Documented in the Portuguese Dictionary of the Lisbon Academy of Sciences since 2001 (Almeida 2008) in three different entries (adjective, adverb, and interjection), the element of surprise or extravagance associated with *bué* has decreased over time as it became increasingly popular among speakers. Currently, such popularity is observed in online forums in which native and non-native speakers of different varieties of Portuguese ask and comment about the words' usage, meaning, and

origins, such as in an entry in the question-and-answer website Quora (Como surgiu o termo "bué" em português de Portugal n.d.) that displays over thirty replies from different perspectives about the word.

One interesting aspect of loanwords that is worth being noted is that it is not clear why speakers borrow a word from a different language when a "fully equivalent word existed beforehand" (Haspelmath 2009, p. 35). And even beyond that, Haspelmath inquires why borrowings occur at all as every language has the means to create new words and expressions. Moreover, while research on loanwords shows that borrowing may take place in both directions—to and from cultures in contact—it is argued that there tends to exist an asymmetry in this process, given that source languages often hold an advantage of power over recipient languages in some way (Hoffer 2002, p. 3). What we observe in the case of *bué* is the opposite, revealing that social aspects and attitudes play a role in borrowing (Poplack 2017, p. 186).

By focusing on the contexts in which the word appears in a collection of contemporary tweets, our analyses showed that this intensifier modifies a variety of word classes, such as adjectives, verbs, and nouns, of both positive and negative semantic evaluation, as well as possesses a quantifier function, suggesting that *bué* displays similar development to other intensifiers crosslinguistically, such as English *lovely* (Adamson 2000) and Spanish and Portuguese *bien/bem* (Kanwit et al. 2017; Lívio and Howe 2020). It starts out as an adjective of positive evaluation, transitioning from a descriptive function to an affective one as speakers use these forms to "externalize their subjective point of view", as argued by Traugott (1999, p. 189). This finding is relevant in the wider literature on intensification as it shows that similar processes of change, such as bleaching and grammaticalization (Ito and Tagliamonte 2003; Luo et al. 2019), are at play in the development of this grammatical class across languages.

In terms of our methods, we demonstrated the usefulness of manipulating social media data, as it provides access to a snapshot of synchrony. This approach allows us to observe how speakers utilize certain terms and constructions, fostering productive dialogues between fields such as language use, pragmatics, historical linguistics, text mining, digital humanities, and computational social science (Huang et al. 2016; Scheffler 2017).

**Author Contributions:** C.L. and C.H. contributed equally to this manuscript. All authors have read and agreed to the published version of the manuscript.

**Funding:** This research received no external funding.

**Institutional Review Board Statement:** Not applicable.

**Informed Consent Statement:** Not applicable.

**Data Availability Statement:** Twitter's current Developer Agreement and Policy prevent us from making the full dataset publicly available.

**Acknowledgments:** The authors would like to express their gratitude to two anonymous reviewers and Patrícia Amaral for providing insightful comments throughout the process. The first author also extends thanks to Carlos Pio for her time and insights regarding the use of the word *bué* in Portugal, along with general comments about language use.

**Conflicts of Interest:** The authors declare no conflicts of interest.

## Notes

[1]  The terms "intensifier" and "booster" are often accompanied by a reference to "maximizers" as well, all with the intention of underscoring the fact that these structures play a role in enhancing the meaning of the modified element. These should not be confused with "minimizers", as in *a word* in "He said not a word", which produces a parallel effect, though in the opposite direction on the degree scale.

[2]  As will be seen in subsequent examples, there are orthographic variants of the canonical intensifiers—e.g., *muito* > *mto* and *mó*. The form *mui* was attested in Old Portuguese (Huber 1986, paras. 423 and 424) and may be observed in contemporary varieties.

[3]　　The publication of *The Routledge Handbook of Spanish Corpus Linguistics* (Parodi et al. 2021) responds to this trend and features recent works on different types of corpora, methodologies, and tools in the field of Spanish linguistics.

[4]　　The code used for the extraction of RAKE returned some strings that were not pairs of ADV+ADJ, such as in *nem precisas*, *já devia*, and *nunca parar*. We manually filtered out those cases and plotted only the ones that matched the search.

[5]　　Among the many features observed in the data provided in Table 1 are, primarily orthographic, strategies used by authors in digital contexts. For instance, in addition to the general use of all caps and repeated letters, laughter in Portuguese is represented specifically as *kkkk/KKKKKK*. Truncated forms such as *tô* (<*estou* 'I am') and *cê* (<*voçe* 'you') are also commonly attested in this register.

[6]　　One of the reviewers points out that this graph may represent two distinct speech communities. We believe that observing the demographic information about these speakers could have been one way to identify such communities.

[7]　　One of the reviewers points out the recurrence of non-Portuguese words, especially in the "word3" slot. We generally concur with Androutsopoulos (2004, p. 83) regarding the use of English words by non-native English speakers in social media. This phenomenon, named 'Englishization' by the author, is described as being motivated by identification with English-speaking pop culture. Furthermore, Androutsopoulos emphasizes the importance of studying Englishization in specific linguistic settings to understand the sociolinguistic factors that may influence such use.

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
