# Peer review of "Text Mining Approaches to Language Use in Social Media: The Case of Portuguese Bué"

_languages, doi:10.3390/languages9030082_

Round 1
Reviewer 1 Report
Comments and Suggestions for Authors
The authors present a clear and important corpus analysis of bué in Angolan Portuguese. Readers walk away from this paper with an understanding of the current variation of a non-English intensifer, which is an important addition to the literature. Below I present some questions and areas for clarification.
1. Can the issue of agency within linguistic change be addressed more directly or explicitly in this paper? The authors use phrasing like "desire to innovate," "draft into service," and cite the maxim of extravagance, leading to an implication of conscious choices in language variation and change.
2. I would like to see the authors add more nuance to the origin of change and further engage with Pimenta's work and the importance of the song. For the reader here, it appears that one song made such an important change. I am not saying it is false by any means, but readers would benefit from an understanding of how that is possible.
3. Lines 88 - 91, I certainly agree with the authors about the competition of the forms and the resulting hierarchy of extravagance. However, I believe it also a good idea to mention how those levels of extravagance will vary according to the speech community. For example, in regions like New England in the US, wicked might not be considered more extravagant than very.
4. The data in Figure 1 is great, but should also be explained more thoroughly in the text. Also, it appears to me that some keyword strings in this plot are not ADV + ADJ combinations, but rather ADV + V combinations.
5. Is the information in Table 2 only for the "bué" spelling?
6. Another note on the spellings. In Figure 2, does the data for the "bué" bar include all possible spellings? If so, I think that just needs to be made explicit.
7. Lines 267 - 268 present a really interesting point. What more can be said about that? For me, there is an important implication about how features are a part of a norm within a speech community and in this case, might we be seeing two different speech communities?
8. Figure 4 - what is the motivation for not collapsing some items into lemmas. One could think that, for example, number marking (boa vs. boas) or spelling differences (estão vs. estao), might not play a significant role in the arguments presented in this section.
9. Figure 4 - how are non-Portuguese entries handled here? (e.g., rough, locked, etc.).
10. Lines 313-316 are critical, but I believe they need to be further developed to help the reader understand the results in the context of the larger question being asked in this paper regarding the distribution of bué. What is it exactly about these words that make them important collocations. Also, please explain to the reader the difference between the significance column and the CorrSignificance column.
11. Discussion & Conclusion - While recognizing that this is a corpus analysis, I find that a mention or brief discussion of language contact and bilingualism is missing. Emphasis is placed on "foreign words" in the corpus but readers will be thinking of this data in terms of everyday speakers. Perhaps the distribution of and collocations around bué are affected by individual or community-level bilingualism.
Reviewer 2 Report
Comments and Suggestions for Authors
I find the article most interesting and well written. I appreciate that the author(s) also look at the phenomenon under study together with the other key varieties of Portuguese. The bibliography cited is extensive. I present my comments and suggestions below:
Line 91: You could also briefly discuss the French intensifier très, based on the Latin trans, which, too, was introduced through semantic change. The interesting point is that moult, the lexical equivalent of, e.g., the Portuguese muito ceased to be used and was replaced by très.
After Figure 4: It would be illustrative to have here one example of the two most frequent collocative combinations. Are there cases in which bué s simultaneously preceded by é and followed by de?
Line 316: Give the table a number and a caption.
Line 470: Is there some special reason for the references not to appear in alphabetical order? For me, at least, this layout is most confusing.
